# Self-restoration of cardiac excitation rhythm by anti-arrhythmic ion channel gating

Rupamanjari Majumder[1†§], Tim De Coster[1,2,3†], Nina Kudryashova[1,2†#], Arie O Verkerk[4,5], Ivan V Kazbanov[2], Balázs Ördög[1], Niels Harlaar[1], Ronald Wilders[4], Antoine AF de Vries[1], Dirk L Ypey[1], Alexander V Panfilov[1,2,6‡], Daniël A Pijnappels[1‡*]

[1]Laboratory of Experimental Cardiology, Department of Cardiology, Leiden University Medical Center, Leiden, Netherlands; [2]Department of Physics and Astronomy, Ghent University, Ghent, Belgium; [3]Department of Cardiovascular Sciences, KU Leuven, Leuven, Belgium; [4]Department of Medical Biology, Amsterdam UMC, Amsterdam, Netherlands; [5]Department of Experimental Cardiology, Amsterdam UMC, Amsterdam, Netherlands; [6]Laboratory of Computational Biology and Medicine, Ural Federal University, Ekaterinburg, Russian Federation

*For correspondence:
D.A.Pijnappels@lumc.nl

[†]These authors contributed equally to this work
[‡]These authors also contributed equally to this work

Present address: [§]Laboratory for Fluid Physics, Pattern Formation and Biocomplexity, Max Planck Institute for Dynamics and Self-Organisation, Göttingen, Germany; [#]Institute for Adaptive and Neural Computation, Informatics Forum, School of Informatics, The University of Edinburgh, Edinburgh, United Kingdom

Competing interests: The authors declare that no competing interests exist.

**Abstract** Homeostatic regulation protects organisms against hazardous physiological changes. However, such regulation is limited in certain organs and associated biological processes. For example, the heart fails to self-restore its normal electrical activity once disturbed, as with sustained arrhythmias. Here we present proof-of-concept of a biological self-restoring system that allows automatic detection and correction of such abnormal excitation rhythms. For the heart, its realization involves the integration of ion channels with newly designed gating properties into cardiomyocytes. This allows cardiac tissue to i) discriminate between normal rhythm and arrhythmia based on frequency-dependent gating and ii) generate an ionic current for termination of the detected arrhythmia. We show in silico, that for both human atrial and ventricular arrhythmias, activation of these channels leads to rapid and repeated restoration of normal excitation rhythm. Experimental validation is provided by injecting the designed channel current for arrhythmia termination in human atrial myocytes using dynamic clamp.

## Introduction

Living organisms strive to maintain a stable internal environment through continuous regulation of the body's physiological processes. Such homeostatic regulation involves three principal mechanisms: detection, processing and effectuation. The 'detector' senses a change in the environment, communicates it to a 'processor', which processes the information and commands the 'effector' to react appropriately (*Horrobin, 1970*). Given their highly conserved nature, these 'detector-effector' (DE) systems are considered to be effective means to respond to changes that, if sustained, may cause harm to the organism. Light sensitivity coupled to movement is an example of a detector-effector mechanism for the full scale of evolved organisms, as exemplified by phototaxis in unicellular organisms (*Nagel et al., 2002*) and pupil reflex in higher order vertebrates (*Sherman and Stark, 1957*). At the cellular level, many DE systems operate through ion channels, which may act as functional detector-effector units. These channels can detect changes in, for example, voltage, hormones or light to which they respond by activation to generate a flow of ions (i.e. an electrical current) or by inactivation to stop a flow of ions (*Catterall, 2000*; *Stojilkovic et al., 2010*; *Govorunova et al.,*

*2015*). Although the human body maintains a multitude of DE systems that contribute to its every day function, a DE system responding specifically to sustained hazardous heart rhythm disturbances seems not to have been evolved. Nevertheless, under natural cardiac electrophysiological conditions, heart rhythm disorders can terminate spontaneously, resulting in non-sustained arrhythmias (*Bub et al., 2003*; *Katritsis et al., 2012*). For those that are sustained, the most effective means for acute restoration of sinus rhythm, especially in atrial and ventricular fibrillation (AF and VF), consists of delivery of high-voltage shocks to the heart for arrhythmia termination (i.e. defibrillation) (*Wellens et al., 1998*; *Link et al., 2010*; *Laslett et al., 2012*; *Borne et al., 2013*; *Kumar and Schwartz, 2014*). This therapy is based on an electronic detector (i.e. sensor of electrical activity) and effector (i.e. electroshock generator), which are incorporated into a single device. Implantation of this device (the implantable cardioverter defibrillator or ICD) allows continuous monitoring of cardiac rhythm in order to ensure rapid arrhythmia detection for automatic delivery of electroshocks. Due to their non-biological nature, these electronic DE systems have a number of shortcomings including limited battery lives, technical malfunction and inappropriate shock deliveries. However, from a patient's perspective, major concerns are the severe pain, anxiety and depression resulting from the electroshocks (*Sears et al., 2011*), not to mention the possibility of permanent tissue damage (*Ashihara and Trayanova, 2005*).

Now imagine that the mammalian heart itself would be able to detect and terminate an arrhythmia in an automatic and repetitive manner just like the ICD, thereby eliminating the need for traumatizing electroshocks. In this article, we present the theoretical fingerprints of a biological DE system that is tailored to detect and terminate cardiac arrhythmias. We refer to it as the Biologically Integrated Cardiac Defibrillator, or BioICD. We envision the BioICD as an ion channel of which the gating is customized to differentiate between sinus rhythm and arrhythmia, and to act accordingly. Conceptually, such an ion channel could be implemented by proper choice of frequency-dependent activation-inactivation kinetics. We use computer modelling to design the kinetics of this channel via three different approaches and demonstrate the possibility for human cardiac tissue to auto-detect and terminate tachyarrhythmias and fibrillation in both atria and ventricles once this type of channel has been integrated. We prove the concept of our study in simulated monolayers of human cardiomyocytes and in anatomically realistic in silico models of the human atrial and ventricular musculature, by showing that in each case considered, the arrhythmia can be detected and terminated to restore sinus rhythm, within seconds of its initiation, on a fully biological basis. In order to provide experimental evidence of the validity of the BioICD concept, we exploited the dynamic clamp technique (*Wilders, 2006*) to realise the designed BioICD function with the use of optogenetically modified human atrial myocytes that are optically stimulated at arrhythmic frequencies.

## Results

Our concept of the BioICD involves the introduction of a new type of ion channel with specific properties into the sarcolemma of cardiac muscle cells, which is possible through genetic modification (*Nyns et al., 2016*). We explored the properties of this ion channel in silico and found that in order to provide the basis for a DE system, such channel should have the ability to sense the frequency of electrical activation through gating mechanisms that are controlled by the cell's membrane potential (voltage difference across the cell membrane). In the subsequent sections, we show the behaviour of a heart endowed with the new ion channel (referred to as 'BioICD'), and compare it with the normal 'Control' situation.

### Principle

In order to show the behaviour of the BioICD ion channel, we first demonstrate how it works in a simplified controllable system, that is tachyarrhythmias in a 2D in silico model consisting exclusively of isotropically arranged human ventricular myocytes. Three possible realisations of this ion channel are provided, each of them relying on a different ionic mechanism: Model I is a minimal model that allows frequency-dependent gating, Model II has a different inactivation mechanism, and Model III has a gating mechanism insensitive to the shape of the action potential (AP). The results of Model II and III can be found in the Supporting Information (Appendix 1 *Figures 1*, *2*, *3*). The detailed description of Model I together with its results is provided below.

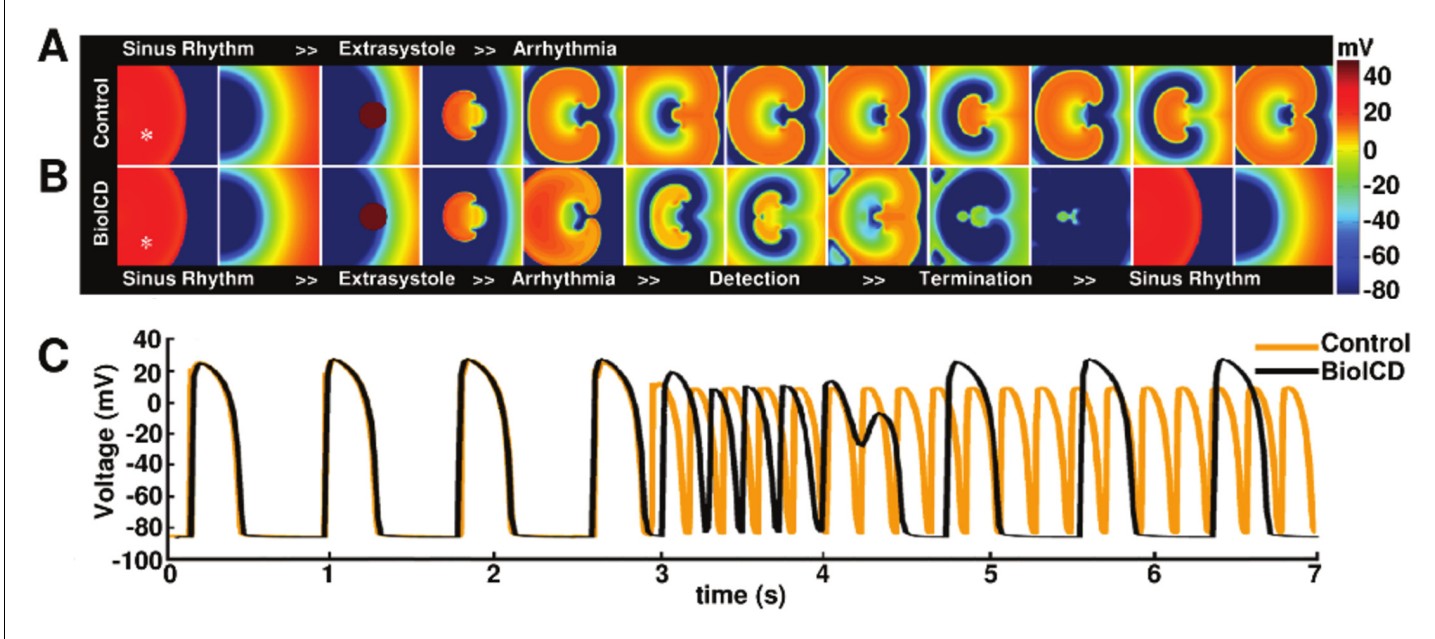

**Figure 1.** Anti-arrhythmic action of the BioICD channel (Model I, see below). (**A**) Development of a sustained tachyarrhythmia, induced via an extrasystole, in a simulated monolayer of normal human ventricular tissue (TNNP model; *ten Tusscher and Panfilov, 2006*). (**B**) Auto-detection and termination of the tachyarrhythmia, followed by restoration of sinus rhythm in the same monolayer upon expression of the BioICD channel. (**C**) Voltage traces recorded from representative cardiomyocytes (white asterisks) in the simulation domains (orange: Control, black: BioICD). Time frames in A-B are chosen to represent the relevant stages in arrhythmia progression, detection and termination and therefore do not correspond linearly with the voltage traces in C.

*Figure 1A* illustrates the control situation for a simulation experiment in a monolayer without expression of the BioICD channel. In this situation, induction of a figure-of-eight reentrant source, via an extrasystole during sinus rhythm, results in a sustained reentrant tachyarrhythmia. In monolayers expressing the BioICD channel (see section *BioICD model I* below), the exact same stimulation first leads to the development of the same reentrant pattern, however then, because of the concurrent increase in activation frequency, the BioICD channel is activated, resulting in termination of the figure-of-eight reentry and restoration of sinus rhythm (*Figure 1B*). Representative voltage traces (*Figure 1C*) demonstrate how the action of the BioICD channel can be recognized in the membrane potential behaviour of a single cardiomyocyte. Of note, the presence of the BioICD channel exerts no influence on wave propagation during sinus rhythm, both before and after the arrhythmia.

## BioICD model I

The schematic representation of Model I is presented in *Figure 2A*. The ion channel comprises a group of subunits that exhibit two states: open ($O$) and closed ($C$), with voltage-dependent transitions between these states, indicated by arrows in *Figure 2A*. The ion channel subunits slowly move to the open state $O$ when voltage ($V$) exceeds $V_{threshold}$, ($V_{threshold}$ is typically around $-60$ mV). When $V < V_{threshold}$, the channel subunits move back to the closed state $C$. Thus, at any given moment in time, the number of open subunits increases with the total time spent by the cardiomyocyte in the suprathreshold (depolarized) state. This ensures that at low activation frequencies the subunits alternate between open and closed states (black line in *Figure 2B*), whereas at high frequencies, temporal summation of activation leads to a higher number of subunits in the open state (black line in *Figure 2C*). We assumed that the ion channel can produce current only if all of its identical subunits are in an open state. Such collective kinetics is required to render the current amplitude sensitive to changes in the activation frequency. In this case, the conductivity of the whole collection of $n$ independent, non-cooperative subunits is proportional to the $n^{th}$ power of $O$ (the fraction of subunits in the open state $O$). Higher values of $n$ provide higher selectivity to the frequency changes, bringing it closer to the all-or-nothing response. Given these conditions, we found that a good intermediate

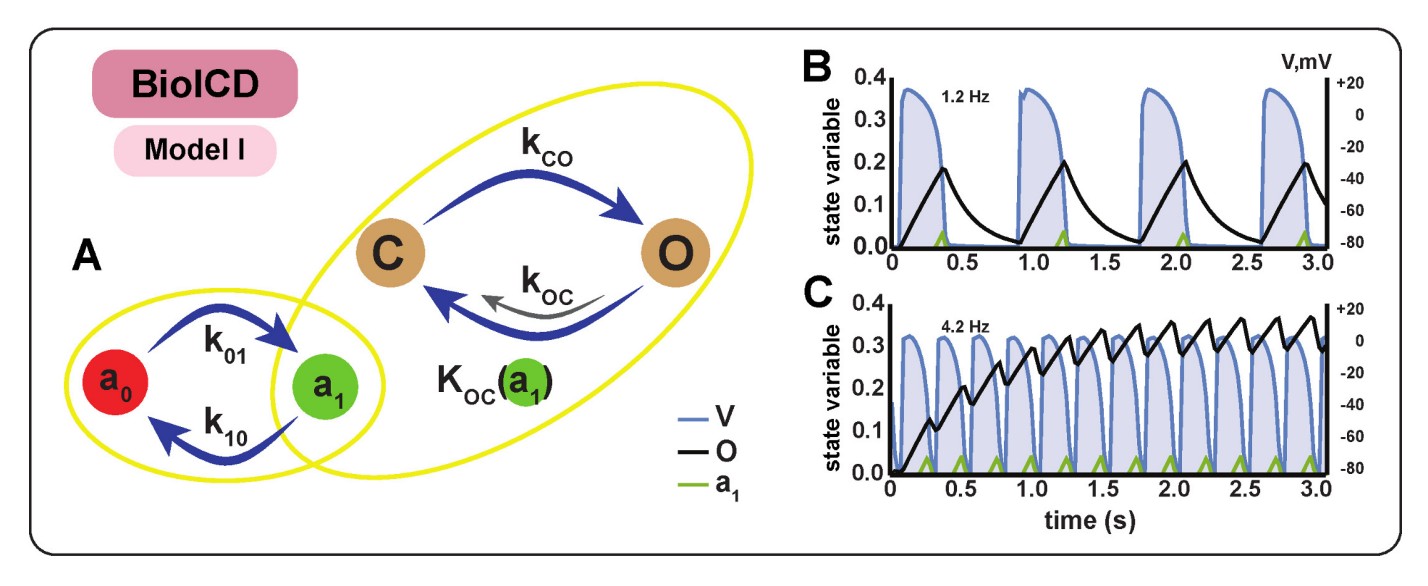

**Figure 2.** Schematic diagram and behaviour of BioICD channel Model I. (A) Markov diagram for BioICD channel Model I. (B) and (C) Traces of voltage (V) and state variables $O$ and $a_1$, at 1.2 Hz and 4.2 Hz, respectively. $O$: open state of the frequency-sensing channel subunit, $C$: closed state of the frequency-sensing channel subunit, $a_0$: inactive state of the catalytic channel deactivation agent, $a_1$: active state of the catalytic channel deactivation agent. A detailed mathematical description of $C$, $O$, $a_0$ and $a_1$ is provided in the Materials and methods section.

solution for effective collective behaviour, which ensures simplicity (low $n$) and accuracy (high $n$), is $n = 8$. Because we require the BioICD channel to produce a depolarizing current upon activation (in line with classical defibrillation), its reversal potential was set close to 0 mV, thus assuming a simple pore without selectivity for any specific ion.

As channel activation occurs at high frequency, it only produces current after the onset of a tachy-arrhythmia (i.e. after a substantial and sustained increase in activation frequency). Activation of this channel raises the membrane potential everywhere in the tissue to $\approx$ 0 mV, and the ion channel sub-units continue to stay in the open state unless they are influenced to move back to the closed state by some separate, yet integrated mechanism. Thus, we included an additional deactivation mecha-nism to restore the initial closed state of the BioICD channel. A 'deactivating agent' $a$ is designed to accumulate in the $a_1$ state if the membrane potential is close to 0 mV for a substantial amount of time. Accumulation of $a_1$ potentiates the deactivation of the BioICD channel (transition from open state $O$ to closed state $C$).

In electrophysiological terms, only if the cell stays depolarized longer than the normal duration of the plateau in an AP, $a_1$ accumulates and facilitates the closure of the BioICD channel. A complete mathematical description and equations of this channel are given in the Materials and methods sec-tion (*Equations 1-6*), and the anti-arrhythmic action of this ion channel at the tissue level is illustrated in *Figure 1*.

## Self-restoration of excitation rhythm in different virtual pathological substrates

Sustained cardiac arrhythmias typically occur in diseased myocardial tissue. Our simulations of human ventricular monolayers with BioICD channels demonstrate that abnormally fast rhythms can be detected and terminated successfully in a variety of pathological substrates. This is illustrated in *Figure 3* (and also in Supporting *Videos 1–4*) for rotor-driven arrhythmias in substrates for (i) ventric-ular tachycardia (VT) (*Figure 3A and E*), (ii) VF (*Figure 3B and F*), (iii) VT in fibrotic tissue (*Figure 3C and G*) and (iv) VT in the presence of an anatomical scar (*Figure 3D and H*). In each case, the BioICD channel activates within 1 s after initiation of the arrhythmias, thereby electrically synchronizing the whole simulation domain within 300 ms and resetting all activity within 700 ms for VT based on func-tional reentry (*Figure 3A and E*), 1000 ms for VF (*Figure 3B and F*) and VT in fibrosis (*Figure 3C*

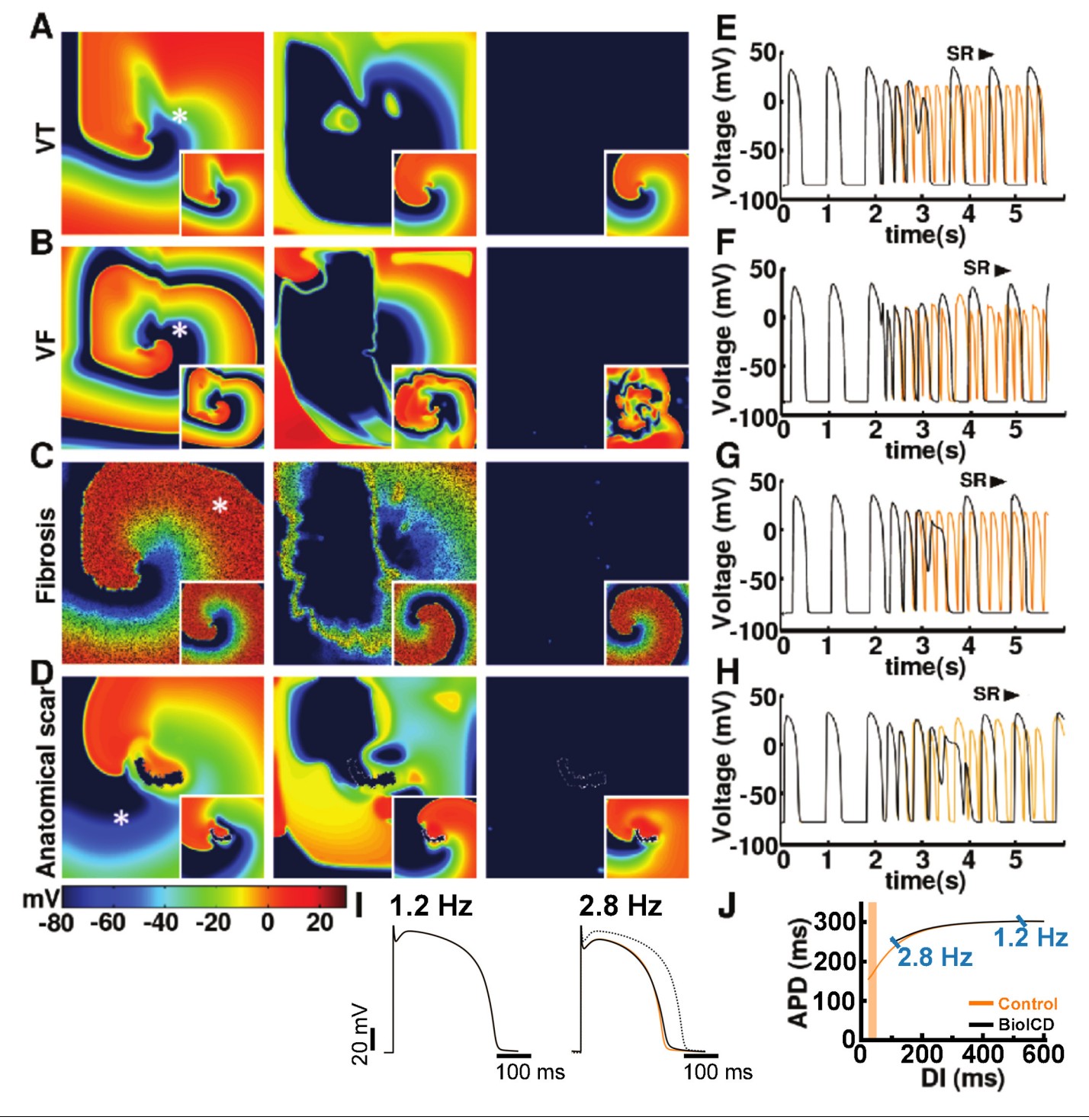

**Figure 3.** Self-restoration of excitation rhythm in human ventricular monolayers with reentrant tachyarrhythmias (TNNP model [***ten Tusscher and Panfilov, 2006***]) for four different pathological substrates. (**A**) A substrate for VT. (**B**) A substrate for VF. (**C**) A substrate with diffuse fibrosis. (**D**) A substrate with an anatomical scar. In each panel, successive frames (from left to right) show the voltage distribution in the monolayers at subsequent relevant time points. The left column shows established reentrant tachyarrhythmias. The middle column shows, for the different arrhythmia conditions, stages of advanced repolarization in the arrhythmia termination process (large dark blue areas), while the right column shows the final tissue-wide repolarization (dark blue) stage, thereby allowing restoration of sinus rhythm (SR). The insets show the corresponding situation in the absence of BioICD channels. (**E–H**) Voltage time series extracted from representative cardiomyocytes (white asterisks) in the simulation domains of situations *A-D*, respectively. (**I**) The BioICD channel has no significant influence on the action potential (AP) at sinus rhythm (1.2 Hz, left AP traces), but slightly increases

*Figure 3 continued on next page*

*Figure 3 continued*

the AP duration in the tail ($APD_{90}$) at close-to-arrhythmic frequencies (right AP traces). The dotted line of the widest AP in the right traces shows the AP at 1.2 Hz for reference. The orange lines show the APs without BioICD current (control). (**J**) APD restitution curve for the original parameters of the human ventricular cardiomyocyte model, with and without BioICD channel. Orange shading is used to indicate the region of the restitution curve where the slope exceeds 1. DI, diastolic interval.

*and G*), and 600 ms for anatomical reentry (*Figure 3D and H*). Sinus rhythm resumes in all cases, once the electrophysiological steady state has been restored. Thus, the entire process from arrhythmia initiation to restoration of sinus rhythm takes about 2 s in the ventricular monolayers. In substrates with more fibrosis (30% instead of 20% fibroblasts as in *Figure 3C and G*), our BioICD model is still able to auto-detect and terminate VT albeit somewhat slower (data not shown).

As shown in *Figure 3I*, the presence of the BioICD channel exerts no significant influence on the AP at sinus rhythm (1.2 Hz) (the black and orange traces coincide), but slightly increases the APD at close-to-arrhythmic frequencies. This is also accounted for in the APD restitution curve of *Figure 3J*, which shows that the BioICD channel effectively increases the minimal APD of the cardiomyocytes. As a consequence of their sensitivity to frequency, the BioICD channels are unable to detect and eliminate reentrant activity anchored to scars if the activation frequency is too low, for example in substrates with large scars (see *Video 5* in the Supporting Information).

## Self-restoration of excitation rhythm in the virtual human heart with sustained fibrillation

Next, we incorporated the BioICD channel in cardiomyocytes of anatomically realistic human atria and ventricles. We find that the same channel successfully detects and terminates sustained fibrillation in both ventricles and atria to restore sinus rhythm, as illustrated in *Figure 4A and B* (the corresponding videos can be found in Supporting Information *Videos 6–7*). The mechanism of defibrillation is similar to that in monolayers. Arrhythmia-triggered activation of the BioICD channel in some regions leads to complete reset of the cells, whereas in others it prolongs APD, leading to extinction of the arrhythmia sources. With our choice of parameters, the BioICD channels in human atria could detect ongoing AF within 1.7 s and terminate it within 1.2 s to restore sinus rhythm. The corresponding channels in the ventricles could detect VF within 3 s and terminate it within 500 ms to restore sinus rhythm.

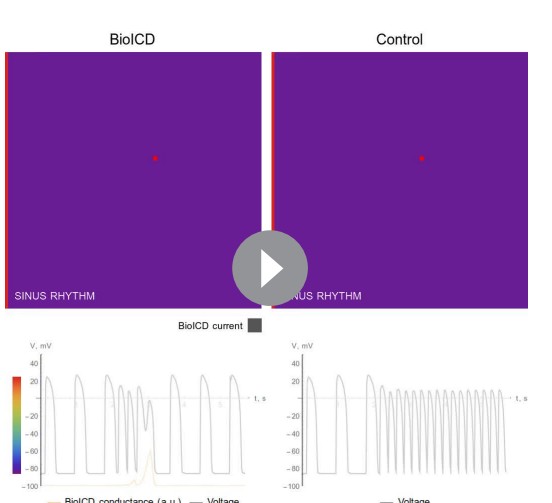

**Video 1.** Self-restoration of cardiac excitation rhythm after VT in monolayers of human ventricular tissue. The activation of the BioICD current is shown in grey, as an overlay on the voltage map in the BioICD case.
https://elifesciences.org/articles/55921#video1

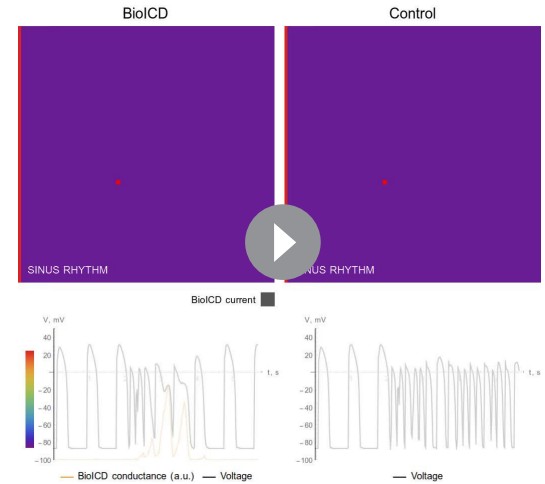

**Video 2.** Self-restoration of cardiac excitation rhythm after VF in monolayers of human ventricular tissue. The activation of the BioICD current is shown in grey, as an overlay on the voltage map in the BioICD case.
https://elifesciences.org/articles/55921#video2

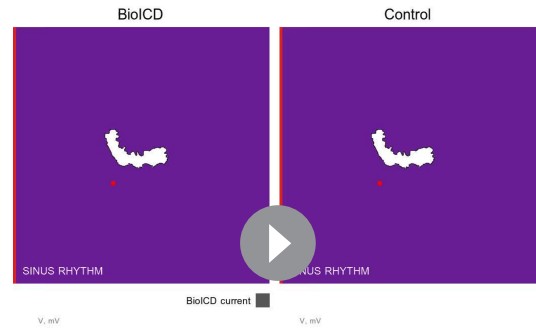

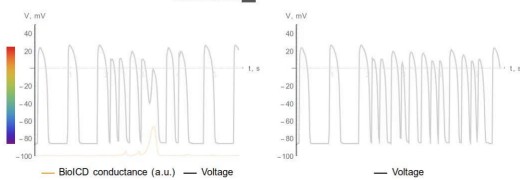

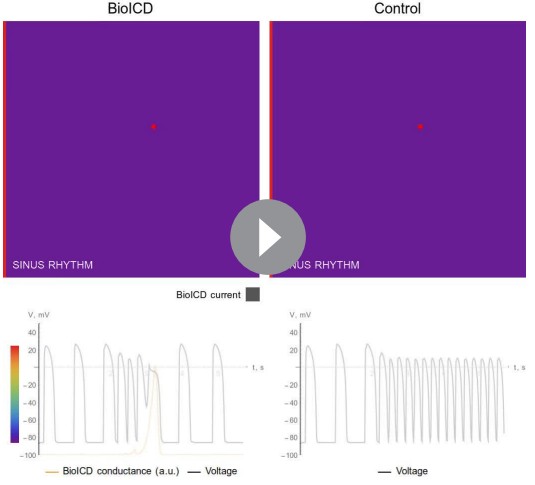

**Video 3.** Self-restoration of cardiac excitation rhythm after VT in monolayers of human ventricular tissue, in the presence of 20% fibroblasts. The activation of the BioICD current is shown in grey, as an overlay on the voltage map in the BioICD case.

https://elifesciences.org/articles/55921#video3

**Video 4.** Self-restoration of cardiac excitation rhythm after VT in monolayers of human ventricular tissue, in the presence of an anatomically realistic scar (3.4 cm long), surrounded by a grey zone, where $I_{Ks}$ was reduced by 80%, $I_{Kr}$ by 70%, $I_{CaL}$ by 69% and $I_{Na}$ by 62%. The activation of the BioICD current is shown in purple, as an overlay on the voltage map in the BioICD case.

https://elifesciences.org/articles/55921#video4

## Self-restoration of excitation rhythm in human atrial myocytes

In order to assess, in actual cardiomyocytes, the anti-arrhythmic effect of the BioICD channel, the BioICD current was mimicked in vitro using dynamic patch-clamping (*Wilders, 2006*). This technique can be used to introduce virtual voltage-gated ion channels (like the BioICD channel) into electrically excitable cells such as cardiomyocytes, in this case human atrial myocytes derived from a conditionally immortalized cell line (*Harlaar et al., 2019*; *Figure 5A*). These cells were optogenetically modified (*Feola et al., 2016*; *Figure 5E*) to control their excitation rhythm by light using a dynamically controlled LED source. The dynamic clamp experiment was carried out in current clamp mode, which allowed the unbiased recording of the membrane potential as it develops as the net result of the activity of all endogenous ion channels and the BioICD channel. The real-time measured membrane potential was used to calculate the amplitude of the BioICD current, which was injected into the cell via the recording electrode. A real-time interface between the patch-clamp amplifier and a computer constituted the feedback loop (*Figure 5B*) that allowed us to mimic the presence of the BioICD channel, and to observe the effects of BioICD channel activity in cardiomyocytes (*Figure 5C–D*).

To simulate arrhythmia conditions, light pacing was switched from 1 Hz to high frequency pacing (*Figure 5C–D*). We found that the same ion channel gating properties used for the in silico studies

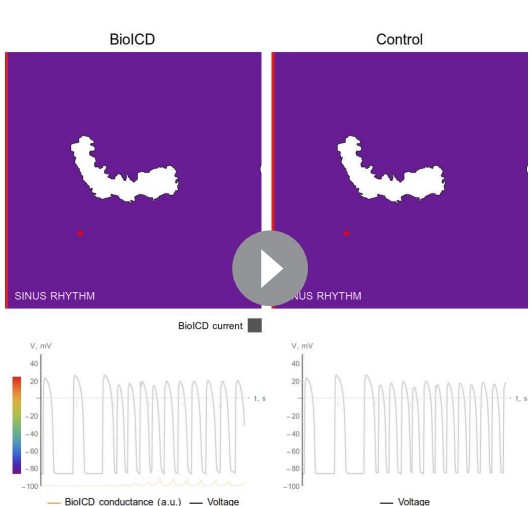

**Video 5.** Failure of self-restoration of cardiac excitation rhythm after VT in monolayers of human ventricular tissue, in the presence of a large anatomically realistic scar (5.8 cm long), surrounded by a grey zone, where $I_{Ks}$ was reduced by 80%, $I_{Kr}$ by 70%, $I_{CaL}$ by 69% and $I_{Na}$ by 62%. The activation of the BioICD current is shown in grey, as an overlay on the voltage map in the BioICD case.

https://elifesciences.org/articles/55921#video5

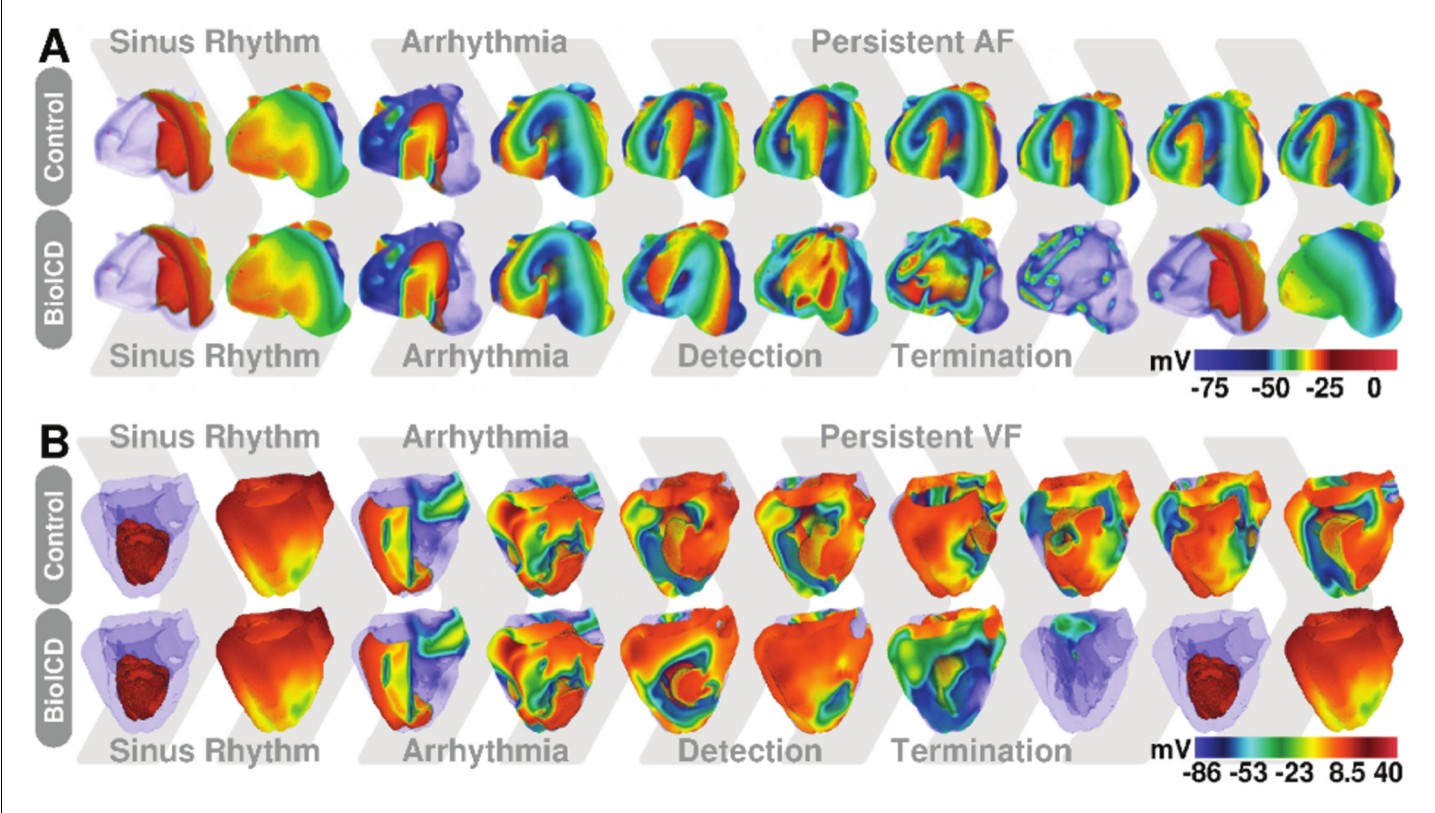

**Figure 4.** Self-restoration of excitation rhythm in anatomically realistic models of human (**A**) atria and (**B**) ventricles with reentrant arrhythmias using BioICD channel Model I (more depolarised regions are more transparent). The upper panels show representative plots of the control situation (in the absence of a BioICD channel). Induction, during sinus rhythm, of reentrant activity by giving an extra external stimulus leads to sustained AF and VF in atria and ventricles, respectively. The lower panels show the exact same situation, but in atria and ventricles expressing the BioICD channel. Once the arrhythmia is induced, the heart itself is able to detect and terminate fibrillation in order to restore sinus rhythm.

successfully detected fast pacing rhythms in a living cell. While no significant distortions of AP were registered at 1 Hz (the black and orange traces coincide), the depolarizing current produced by the BioICD channel at high frequencies built up resulting in APD prolongation. In tissue, the cells depolarised by the BioICD current would create conduction block to 'defibrillate'. In order to simulate this tissue-level feedback in a single-cell set-up, the light-pacing protocol was augmented with a feedback mechanism that detected pacing block (simulating conduction block in tissue) and subsequently lowered the pacing frequency, leading to restoration of the original cardiac excitation rhythm. *Figure 5F* shows a zoom-in of panel 5C focussing on the termination phase.

This process of arrhythmia detection, current accumulation, and pacing block is frequency-dependent, as evidenced by the smaller time until termination following 8 Hz optical pacing (1.16 s, *Figure 5D*) versus 7 Hz pacing (1.92 s, *Figure 5C*) within the same cell. The frequency at which termination occurs is also dependent on the baseline APD (*Figure 5G*). For a total of 7 cells showing different baseline APDs (APD at 90% of repolarization ($APD_{90}$) measured at 1 Hz), basic cycle length (BCL) was decreased slowly (frequency was increased) to find the maximal BCL at which self-restoration of excitation rhythm would occur. Upon an increase in baseline APD, the BioICD channel is activated earlier, namely at longer BCLs. Collectively, these findings show that the proposed ionic gating mechanism is able to restore cardiac excitation rhythm in living excitable cells.

## Discussion

In this study we present evidence that the heart itself may be empowered to detect and terminate arrhythmias. This is achieved by adding only one additional type of customized ion channel, with tailored frequency-dependent gating properties, to the existing repertoire of ion channels. In virtual

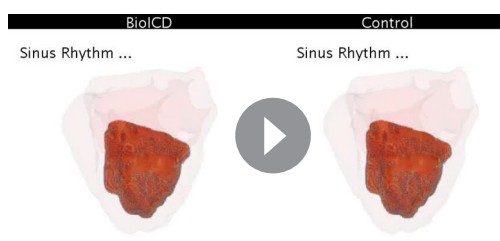

**Video 6.** Self-restoration of cardiac excitation rhythm after VF in anatomically realistic human ventricles.
https://elifesciences.org/articles/55921#video6

hearts equipped with this so-called BioICD channel, sinus rhythm is rapidly restored after arrhythmia initiation, in a fully automatic and shock-free manner without the need of a non-biological factor.

Regarding the customized gating properties of the BioICD ion channel, we show that, amongst the parameters characteristic for tachyarrhythmia or fibrillation, even an obvious one like its frequency can already provide a basis for the detection mechanism. Here, arrhythmia detection relies on positive feedback upon frequency increase. We propose three possible realisations of such anti-arrhythmic gating, as described in the different BioICD models I-III. However, these should not be viewed as the ultimate designs of a BioICD channel, but more as a first set of rigorous outlines that can serve as a basis for additional research now that the concept has been proven in the virtual human heart. Our model can be further improved or modified in several aspects (see Supplementary Information), for example the power $n$ can be lowered. In proteins, this can result in a good trade-off between the detecting-terminating abilities of the BioICD ion channels and their complexity. Also, changing time constants will alter the activation time course of the channel, which can result in a shorter or longer delay preceding arrhythmia termination. Indeed, the structure of our BioICD model allows ample freedom to choose kinetic as well as threshold parameters for the channel within certain limits. As proof-of-principle in the virtual human heart, we tested the BioICD channels in homogeneous, anatomically realistic atria and ventricles. Based on our dynamic patch-clamp experiments, we conjecture that the model should also work for pathological substrates where the arrhythmia comes with an increase in activation frequency. This is based on the notion that the mechanism for self-restoration of cardiac excitation rhythm relies only on the detection of high-frequency signals, irrespective of the nature and underlying cause of the arrhythmia.

Having explained the guiding principles behind the functioning of BioICD channels at the cellular level, we find it fascinating to consider the resemblance between functioning of these ion channels at the organ level and 'spontaneous termination' of arrhythmias. Note that, in nature, not all high-frequency arrhythmias are sustained. A significant fraction does get self-terminated (*Josephson and Kastor, 1977*), although the mechanisms behind such spontaneous termination are still up for speculation. For example, reentrant circuits can act as feedback systems in which changes in the duration of one cycle have the potential to affect conduction in the subsequent cycles (*Frame and Rhee, 1991*). Cycle length was found to oscillate with progressively increasing alternations of long and short periods until a sufficiently short cycle led to conduction block (*Ortiz et al., 1993*; *Frame and Simson, 1988*; *Quan and Rudy, 1990*). A feedback system with the aforementioned characteristic was studied in silico and in vivo in the atrioventricular-node (*Simson et al., 1981*; *Sun et al., 1995*). Also, a frequency-based feedback system exists in the atrioventricular and sinoatrial nodes in the heart, where overdrive suppression creates block of nodal activation (*Kunysz et al., 1995*; *Vassalle, 1970*; *Kunysz et al., 1997*). In addition, *Nagai et al., 2000* demonstrated a way to end paroxysmal arrhythmias based on overdrive suppression in a circular ring of regular cardiac cells interspersed with two pacemakers. Interestingly, whereas unstable tachycardias demonstrate cycle length oscillations that can lead to spontaneous termination of the tachycardia, stable tachycardias exhibit damped oscillations when perturbed. Despite the phenomena discussed above, these damped oscillations can still lead to sustained reentry. In the present study, we focused on developing a DE system capable of handling

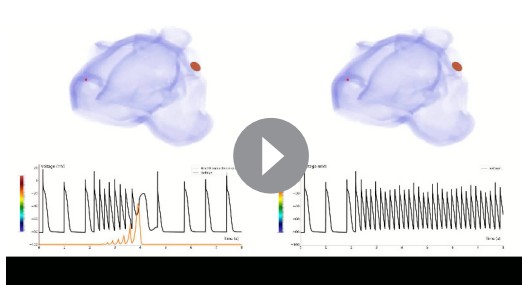

**Video 7.** Self-restoration of cardiac excitation rhythm after AF in anatomically realistic human atria.
https://elifesciences.org/articles/55921#video7

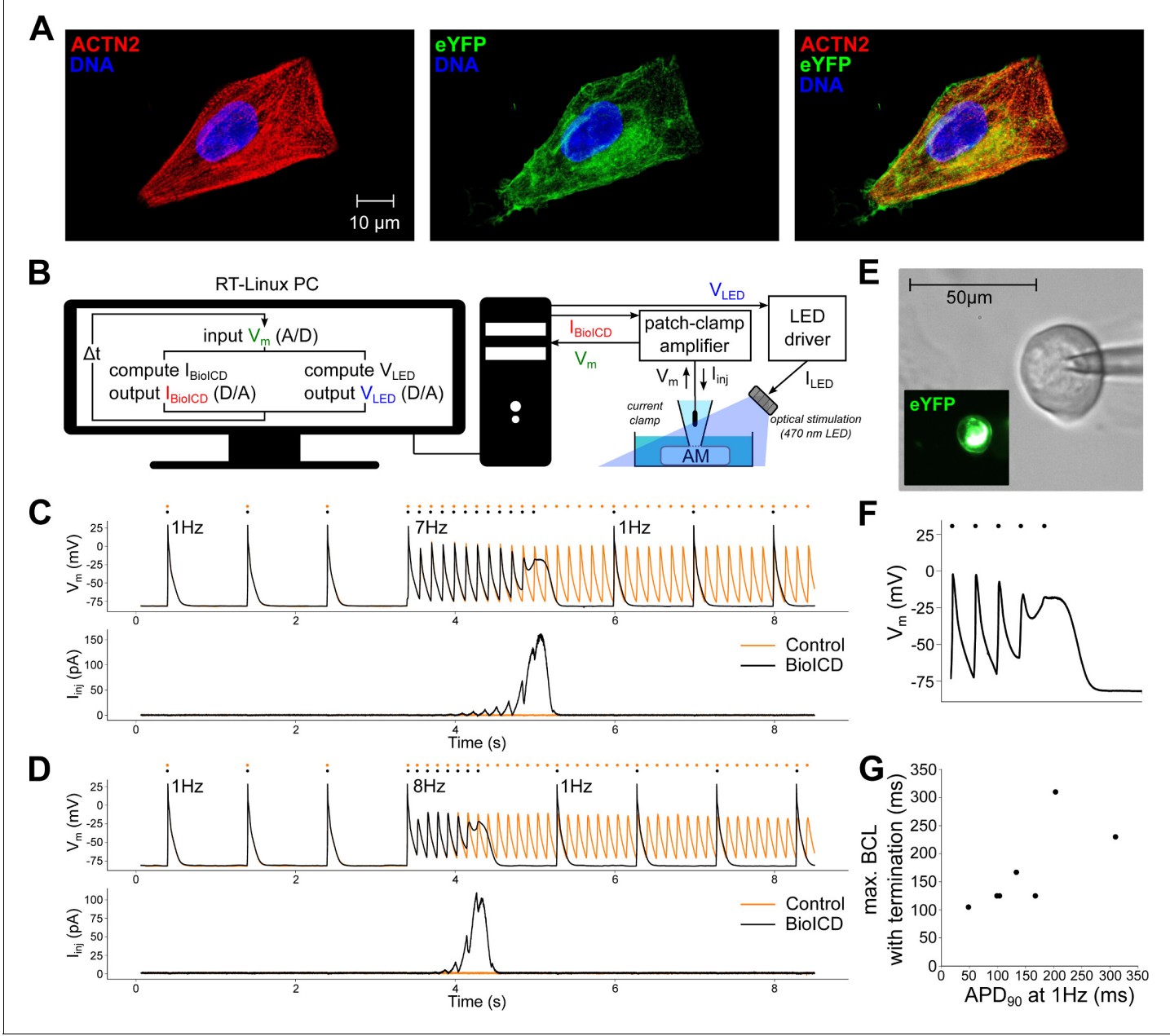

**Figure 5.** Self-restoration of excitation rhythm in cultured human atrial myocytes (AM) with the help of dynamic patch-clamping using BioICD channel Model I. (**A**) Human atrial myocytes stained for (from left to right) α-actinin (ACTN2), enhanced yellow fluorescent protein (eYFP) and DNA. (**B**) Dynamic patch-clamp set-up creating a feedback system for current injection and optical pacing. (**C**) Typical membrane potential ($V_m$) and injected current ($I_{inj}$) traces with and without the BioICD current ($I_{BioICD}$) enabled at 7 Hz. Both traces are from the same cell and were recorded 7 s apart. Dots indicate the stimulus times. (**D**) Typical membrane potential and injected current traces with and without the BioICD current enabled in the same cell as (**C**) at 8 Hz. Also here, both traces were recorded 7 s apart. (**E**) eYFP fluorescence produced by optogenetically modified human atrial myocyte. (**F**) Zoom-in of the trace in (**C**) showing termination of arrhythmic activity. The last optical stimulus is blocked, allowing 1 Hz activation to regain. (**G**) Maximal basic cycle length (BCL) for which termination still occurs as a function of baseline APD. Data obtained from 7 cells. Other abbreviations: real-time (RT), analog/digital (A/D), voltage input for the LED driver ($V_{LED}$), current output of the LED driver ($I_{LED}$).

The online version of this article includes the following source data for figure 5:

**Source data 1.** Dynamic clamp reading 1: CSV-file containing the raw data for the C-panel in *Figure 5*.
**Source data 2.** Dynamic clamp reading 2: CSV-file containing the raw data for the D-panel in *Figure 5*.
**Source data 3.** Dynamic clamp summary table: Table containing the experimental datapoints that were plotted in *Figure 5G*.

sustained arrhythmias, including the most hazardous ones like fibrillation. This concept came to expression in the design of anti-arrhythmic ion channel gating, that is the BioICD channel.

As a consequence of the design, this particular BioICD model allows for the detection and elimination of relatively fast sustained rhythms, thereby targeting those arrhythmias that require intervention for acute termination. This means that relatively low frequency arrhythmias are not detected by the BioICD channel and are therefore also not terminated. An example is a reentrant wave anchored to an anatomical obstacle (i.e. a scar) with a large perimeter (see Supporting Information, *Video 5*). In this case, the period of rotation, which is determined by the perimeter of the scar and directly related to the activation frequency, stays below the detection threshold. Thus, the abnormal electrical activity sustains in the cardiac tissue, despite the presence of the BioICD channel. However, as low-frequency arrhythmias are considered to be less hazardous than those with a high frequency, like fibrillation, they can be managed otherwise (*Cho and Marbán, 2010*; *Stevenson, 2013*; *Nattel et al., 2014*; *Bongianino and Priori, 2015*). Alternatively, other biological arrhythmia detection mechanisms could be explored and developed if needed.

Additional next steps in developing the BioICD channel, from a computational point of view, should be (*i*) extending the space of synthetic models, together with (*ii*) modifying existing in silico models of known ion channels to propose a candidate for experimental realization. Indeed, future research should lead to identification of the most robust and broadly applicable BioICD channel with the highest translational potential. We believe that, for the construction of a BioICD channel, all the essential knowledge about ion channel structure-function relationships (*Börjesson and Elinder, 2008*; *Catterall, 2010*; *Moreau et al., 2014*), as well as the molecular tools to manipulate these relationships in a systemic and rational manner are available (*Bayley and Jayasinghe, 2004*; *Subramanyam and Colecraft, 2015*). The ion channels from which functions can serve as a blueprint are not limited to eukaryotic organisms only, but encompass prokaryotic ones as well (*Ren et al., 2001*; *Martinac et al., 2008*) and can be searched for in extensive databases, such as the IUPHAR/BPS (*Southan et al., 2016*). Also, the field of optogenetics has allowed the study of biological arrhythmia termination and could thereby add to the translation of our findings (*Boyle et al., 2018*; *Entcheva and Bub, 2016*; *Boyden et al., 2005*). Here, cardiomyocytes are genetically modified to express light-gated ion channels, allowing control of electric current generation in these cells by light. In earlier work from our group and others, it was shown that cardiomyocytes are indeed able to generate sufficient electrical current, based on the natural electrochemical gradients, for biological auto-termination of both atrial and ventricular arrhythmias, including fibrillation. This was shown in vitro using monolayers of neonatal atrial myocytes (*Bingen et al., 2014*), but later also in the whole heart of adult rodents (*Bruegmann et al., 2016*; *Crocini et al., 2016*; *Nyns et al., 2016*), and most recently also integrated in a hybrid bio-electronic system (*Nyns et al., 2019*). In these studies, the gating of ion channels was controlled by light to terminate cardiac arrhythmias. As a next step, we now show how such gating could be controlled by the arrhythmia itself, thereby giving rise to a new fully biological DE system.

In this study, we used optogenetics to provide experimental proof of the aforementioned concept with the use of human atrial myocytes. These atrial myocytes were optogenetically modified and used in dynamic patch-clamp experiments to separate the pacing mechanism (optical) from the rhythm restoration mechanism (chemical/electrical) in order to improve signal interpretation, that is stimulation block at depolarised membrane potentials. Our studies reveal that the in vitro data (*Figure 5C–D*) show the same phenomenological phases as visible in the traces coming from the in silico simulations (*Figure 1*). Because the BioICD channel is capable of restoring normal excitation rhythm in cells with short baseline APDs under high frequency, the design of the gating properties makes that such restoration will work also for cells with long baseline APDs (*Figures 2* and *5G*), like those found in ventricular myocytes. This motivated our decision to use an atrial cell line for the experimental validation of our proposed concept.

Taken together, our study presents insight into acquired homeostatic regulation of excitation rhythm under disturbed conditions, including cardiac fibrillation. It is shown, by a combination of theoretical and experimental studies, that such regulation can be established by creation of a biologically engineered DE system for arrhythmias through expression of customized ion channels. Self-resetting of an acutely disturbed heart rhythm by an engineered biological DE system may yield unique insight into arrhythmia management and may lay the foundation for the development of distinctively innovative treatment options by the creation of new biology for therapeutic purposes,

ultimately leading to acute, yet trauma-free termination of arrhythmias. This would stretch the field of synthetic biology into cardiology and also provide a radically new perspective for other medical fields, given the general nature and versatility of biological DE systems. This perspective involves that a diseased organ, begets its own remedy, for example a Biologically Integrated Cardiac Defibrillator (BioICD) in case of the heart.

# Materials and methods

## Key resources table

| Reagent type (species) or resource | Designation | Source or reference | Identifiers | Additional information |
|---|---|---|---|---|
| Cell line (*Homo-sapiens*) | Conditionally immortalised human atrial cell line | DOI: 10.1093/eurheartj | | |
| Gene (*Chlamydomonas reinhardtii*) | ChR2(H134R) | Addgene | plasmid #26973 | source of the H134R variant of channelrhodopsin 2 |
| Genetic reagent | LVV-ChR2(H134R) | This paper | | Lentiviral vector particles to transduce and express ChR2(H134R) |
| Software, algorithm | graphics processing unit-usable code | DOI: 10.1152/ajp-heart.00109.2006, DOI: 10.1038/srep20835 | | |
| Software, algorithm | RTXI dynamic clamp software | DOI: 10.1371/journal. pcbi.1005430 | | |

Detailed formulations for our BioICD ion channel model, based on a Markov chain formalism (*Hermanns, 2002*; *Rudy and Silva, 2006*), are provided below.

## BioICD model I

Total BioICD current is expressed as:

$$I_{BioICD} = G_{BioICD}O^8(V - E_{BioICD}). \tag{1}$$

where $G_{BioICD} = 150\,\text{nS/pF}$ is the maximal conductance, and $E_{BioICD} = 0\,\text{mV}$ denotes the reversal potential of the BioICD channel. There is a frequency-sensing channel subunit with 2 states: open (O) and closed (C). Eight subunits together compose a BioICD channel. Deactivation of the channel is facilitated with a catalytic agent ($a$), which has 2 states: inactive $a_0$ and active $a_1$. These states evolve according to *Equations 2, 3, 4, 5, 6*:

$$\frac{\partial O}{\partial t} = \frac{k_{CO}C - k'_{OC}(a_1)O}{\tau_o}, \tag{2}$$

$$\frac{\partial a_1}{\partial t} = \frac{k_{01}a_0 - k_{10}a_1}{\tau_a}, \tag{3}$$

$$k'_{OC}(a_1) = k_{OC} + K_{OC}(a_1), \tag{4}$$

$$C + O = 1 \tag{5}$$

$$a_0 + a_1 = 1 \tag{6}$$

where $\tau_o = \tau_a = 1400\,\text{ms}$ is the characteristic time, $K_{OC}(a_1) = 2 \cdot 10^4 a_1{}^5$, and $k_{CO}$, $k_{OC}$, $k_{01}$, and $k_{10}$ are functions of voltage that determine the opening and closing rates:

$$k_{CO} = \begin{cases} 0.0 & \text{if } V < -60\,\text{mV}, \\ 1.0 & \text{otherwise}, \end{cases} \tag{7}$$

$$k_{OC} = \begin{cases} 7.0 & \text{if } V < -60\,\text{mV}, \\ 0.0 & \text{otherwise}, \end{cases} \tag{8}$$

$$k_{01} = \begin{cases} 1.0 & \text{if } -55\,\text{mV} < V < 0\,\text{mV}, \\ 0.0 & \text{otherwise}, \end{cases} \tag{9}$$

$$k_{10} = \begin{cases} 0.0 & \text{if } -55\,\text{mV} < V < 0\,\text{mV}, \\ 200.0 & \text{otherwise}. \end{cases} \tag{10}$$

Here, we assume that the number of catalyst molecules in the cell is much higher than the number of BioICD channels, such that the binding between them does not significantly change the number of active catalysts $a_1$. The fifth power of $a_1$ in $K_{OC}(a_1)$ stands for the non-cooperative binding of 5 catalyst molecules to the BioICD channel. The full detailed Markov chain model for this process requires a large number of catalyst-bound states. This type of models can often be reduced to lower dimensions, as described by *Keener, 2009*. We thus substituted the complex catalyst-binding process with an extra facilitated flow from state $O$ to state $C$. The reduced model reproduces the same overall kinetics of the $O \rightarrow C$ catalysed reaction and is more suitable for whole-heart simulations due to its simplicity.

## Human cardiac tissue model

To study the effect of the BioICD channel in human cardiac tissue, we integrated the BioICD channel model as a conductance into the electrophysiological model of each cardiomyocyte. For human ventricular tissue, the mathematical model proposed by *Ten Tusscher et al., 2009* was used, where the transmembrane voltage ($V$) is calculated in millivolts (mV) according to *Equation 11*:

$$\frac{\partial V}{\partial t} = \nabla\left(\tilde{D}\nabla V\right) - \frac{I_{ion} + I_{stim}}{C_m}, \tag{11}$$

where $t$ is time in milliseconds (ms), $I_{ion}$ is the total ionic current density in micro-amperes per square centimetre ($\mu$A/cm$^2$), $I_{stim}$ is the external stimulus current, $C_m$ is the specific membrane capacitance in microfarad per square centimetre ($\mu$F/cm$^2$), and $\tilde{D}$ is the diffusion tensor, whose components are related to the electrical conductivity of cardiac tissue in each direction of propagation.

$$\begin{aligned} I_{ion} &= I_{Na} + I_{CaL} + I_{K1} + I_{to} + I_{NaCa} \\ &+ I_{NaK} + I_{Kr} + I_{Ks} + I_{bNa} + I_{bCa} + I_{pCa} + I_{pK} \\ &+ I_{BioICD}. \end{aligned} \tag{12}$$

Here, the different cardiac currents are represented as follows: $I_{Na}$: the fast $Na^+$ current, $I_{CaL}$: the L-type $Ca^{2+}$ current, $I_{K1}$: the inward-rectifier $K^+$ current, $I_{to}$: the transient outward $K^+$ current, $I_{NaCa}$: the $Na^+/Ca^{2+}$ exchanger current, $I_{NaK}$: the $Na^+/K^+$ pump current, $I_{Kr}$: the rapid delayed rectifier $K^+$ current, $I_{Ks}$: the slow delayed rectifier $K^+$ current, $I_{bNa}$: the background $Na^+$ current, $I_{bCa}$: the background $Ca^{2+}$ current, $I_{pCa}$: the plateau $Ca^{2+}$ current, and $I_{pK}$: the plateau $K^+$ current. Units for conductance measurements ($G_X$) and measurements of intracellular and extracellular ionic concentrations ($[X]_i$ and $[X]_o$), are in nanosiemens per picofarad (nS/pF) and millimole per litre (mM), respectively.

For simulations of *Figure 3*, we used a monolayer model, containing $512 \times 512$ grid points, such that the physical size of the simulated tissue was $12.8 \times 12.8$ cm. Our model for fibrosis contained 20% randomly distributed fibroblasts, which did not couple electrotonically to the neighbouring cardiomyocytes (*Ten Tusscher and Panfilov, 2007*). For simulations with the realistic scar, a grey border zone (GZ) was taken into consideration. In the GZ, $I_{Ks}$ was reduced by 80%, $I_{Kr}$ by 70%, $I_{CaL}$ by 69% and $I_{Na}$ by 62% (*Arevalo et al., 2013*).

For human atrial tissue, data and fibre directions were obtained from *Dössel et al., 2011*. The ionic cell model that was used is the AF-induced electrically remodelled human atrial *Courtemanche et al., 1999* model (adapted from *Courtemanche et al., 1998*, which was taken from CellML (*Yu et al., 2011*) with the use of Myokit (*Clerx et al., 2016*) and transformed to

graphics processing unit-usable code). The transmembrane voltage ($V$) is described in millivolts (mV) according to *Equation 12* with $I_{pK}$ replaced by $I_{Kur}$, an ultra-rapid delayed rectifier $K^+$ current.

We used an S1-S2 cross-field protocol to generate spiral and scroll waves (*ten Tusscher and Panfilov, 2006*). In human ventricular monolayers, we assumed isotropy. Thus, the diffusion tensor of *Equation 11* could be replaced by a scalar coefficient D. However, in 3D, owing to the natural anisotropy of realistic cardiac tissue, the elements were computed on the basis of a reconstructed fibre direction field as described by *Ten Tusscher et al., 2009*. The transverse diffusion coefficient ($D_t$, for signal propagation across the fibers) was assumed to be 4 (ventricles) or 6 (atria) times less than the longitudinal diffusion coefficient ($D_l$, for signal propagation along the fibers). Elements of the diffusion tensor were computed as follows:

$$D_{ij} = (D_l - D_t)\alpha_i\alpha_j + D_t\delta_{ij}, \tag{13}$$

where $\alpha_i$ are components of the unit vector that is oriented along the direction of a fibre. We used $D_l = 1.54 \, \text{cm}^2/\text{s}$ for ventricles and $D_l = 1.6 \, \text{cm}^2/\text{s}$ for atria.

We integrated *Equation 11* in time using the forward Euler method with time step $\delta t = 0.02 \, \text{ms}$ (ventricular model) or $\delta t = 0.01 \, \text{ms}$ (atrial model), and in space, using the centred finite-differencing scheme with space step $\delta x = \delta y = 0.025 \, \text{cm}$ in monolayers, 0.05 cm in whole ventricles and 0.0330 cm in whole atria, subject to 'no flux' boundary conditions. The simulation domains for the whole ventricles and atria contained 1, 693, 010 and 2, 173, 891 grid points, respectively. The gating variables in the electrophysiological model for the human cardiomyocyte were integrated using the *Rush and Larsen, 1978* scheme.

## Cell lines

The cell line used in this study has recently been generated by transducing human foetal atrial myocytes with a human immunodeficiency virus type 1-based vector conferring doxycycline-controlled expression of simian virus 40 large tumour antigen as described in *Harlaar et al., 2019*. The cells were tested negative for the presence of mycoplasma by the MycoAlert mycoplasma detection kit (Lonza, Basel, Switzerland), as well as by Hoechst 33342 (Life Technologies Europe, Bleiswijk, the Netherlands) staining of Vero cells cultured with conditioned human atrial myocyte medium.

## Preparation of ChR2(H134R)-expressing human atrial myocytes

Self-inactivating lentiviral vector particles encoding an eYFP-tagged version of the H134R mutant of *Chlamydomonas reinhardtii* channelrhodopsin 2 (LVV-ChR2(H134R)), were created by standard laboratory procedures (*Feola et al., 2016*) and used to transduce human atrial myocytes differentiated from a conditionally immortalised cell line (*Harlaar et al., 2019*). On the day before the patch-clamp experiments, cells were dissociated and re-plated on fibronectin-coated glass coverslips at $2.5 \times 10^4$ cells per well in 24-well culture plates (Greiner Bio-One, Alphen aan den Rijn, the Netherlands).

## Electrophysiological analysis

APs were recorded in single cells or small clusters of cells using the amphotericin-B perforated patch-clamp technique (*Wilders, 2006*). Voltage control, data acquisition, and analysis were accomplished with custom-made software. Potentials were corrected for the calculated liquid junction potential (*Barry and Lynch, 1991*). Signals were low-pass-filtered (cut-off of 5 kHz) and digitized at 5 kHz. Cell membrane capacitance ($C_m$) was estimated by dividing the time constant of the decay of the capacitive transient in response to 5 mV hyperpolarizing voltage clamp steps from $-40$ mV by the series resistance.

APs were measured at $36 \pm 0.2°$C in extracellular solution containing (in mM): *NaCl* 140, *KCl* 5.4, *CaCl*$_2$ 1.8, *MgCl*$_2$ 1, glucose 5.5, HEPES 5; pH 7.4 (*NaOH*). Pipettes (resistance $2.5 - 3$ M$\Omega$) were pulled from borosilicate glass capillaries (Harvard Apparatus, Cambourne, UK) using a custom-made microelectrode puller, and filled with solution containing (in mM): K-gluconate 125, *KCl* 20, *NaCl* 5, amphotericin-B 0.44, HEPES 10; pH 7.2 (*KOH*). APs were elicited by 10 ms pulses of light (470 nm, 3mW/mm$^2$). The light source comprised of a 470L3-C4 mounted LED (Thorlabs, Newton, NJ), driven by the LEDD1B modulated power supply. Light was delivered via the COP1-A collimation lens assembly (Thorlabs, Newton, NJ) positioned above the bath at a distance of 15 cm.

## Dynamic clamp protocol

A Linux computer running RTXI dynamic clamp software (*Patel et al., 2017*) with a time step $\Delta t = 50\ \mu\text{s}$ allowed real-time dynamic interaction between the input and output channels of the patch-clamp amplifier. At a variety of frequencies, an optical stimulus train was applied that consisted of the following phases: i) 7 pulses of pre-pacing are applied at 1 Hz with the BioICD channel enabled, ii) 8 pulses are applied at 1 Hz with the BioICD channel still enabled, iii) an undetermined number of pulses is applied at increased frequency with the BioICD channel enabled, either until termination occurs due to the feedback mechanism, or until a stable state has been achieved (in which case the protocol is terminated), iv) after restoration of excitation rhythm, another 7 pulses are applied at 1 Hz with the BioICD channel enabled, v) 8 pulses are applied at 1 Hz with the BioICD channel disabled, vi) an undetermined number of pulses is applied at increased frequency with the BioICD channel still disabled until the stimulus train is stopped externally. The feedback mechanism detected refractory beats as defined by membrane potential values more positive than $-20$ mV during the entire pulse duration.

## Data and software availability

The key component of the graphics processing unit-usable code that is sufficient for reproduction of the results was made readily available. More detailed information about the data and software that support the findings of this study are available from the corresponding author upon reasonable request.

## Acknowledgements

This study was supported by The European Research Council (ERC Starting grant 716509) to DAP. Additional support was provided by the Netherlands Organisation for Scientific Research (NWO Vidi grant 91714336) to DAP, and by Ammodo (to DAP and AAFdV). The line of conditionally immortalised human atrial myocytes used in this study was made with financial support of the research programme More Knowledge with Fewer Animals (MKMD) with project number 114022503 (to AAFdV), which is (partly) financed by the Netherlands Organisation for Health Research and Development (ZonMw) and the Dutch Society for the Replacement of Animal Testing (dsRAT), and of the Leiden Regenerative Medicine Platform Holding (LRMPH project 8212/41235 to AAFdeV). We would like to thank Marie-José Goumans and Tessa van Herwaarden for their involvement in human tissue handling, Prof. Gunnar Seemann for helping with the atrial modeling, and Prof. Leon Glass and Alexander S Teplenin for useful discussions.

## Additional information

### Funding

| Funder | Grant reference number | Author |
| --- | --- | --- |
| European Research Council | ERC starting grant 716509 | Daniël A Pijnappels |
| Netherlands Organisation for Scientific Research | NWO Vidi grant 91714336 | Daniël A Pijnappels |
| Ammodo Foundation | | Daniël A Pijnappels Antoine AF de Vries |
| Netherlands Organisation for Health Research and Development | project 114022503 | Antoine AF de Vries |
| Leiden Regenerative Medicine Platform Holding | LRMPH project 8212/41235 | Antoine AF de Vries |
| Dutch Society for the Replacement of Animal Testing | | Antoine AF de Vries |

The funders had no role in study design, data collection and interpretation, or the decision to submit the work for publication.

## Author contributions
Rupamanjari Majumder, Tim De Coster, Nina Kudryashova, Conceptualization, Data curation, Software, Formal analysis, Validation, Investigation, Visualization, Methodology, Writing - original draft, Writing - review and editing; Arie O Verkerk, Conceptualization, Data curation, Formal analysis, Investigation, Visualization, Methodology, Writing - original draft, Writing - review and editing; Ivan V Kazbanov, Conceptualization, Software, Investigation, Writing - original draft, Writing - review and editing; Balázs Ördög, Conceptualization, Formal analysis, Validation, Investigation, Methodology, Writing - original draft, Writing - review and editing; Niels Harlaar, Investigation, Visualization, Methodology, Writing - original draft, Writing - review and editing; Ronald Wilders, Conceptualization, Software, Investigation, Methodology, Writing - original draft, Writing - review and editing; Antoine AF de Vries, Conceptualization, Resources, Funding acquisition, Methodology, Writing - original draft, Writing - review and editing; Dirk L Ypey, Conceptualization, Supervision, Validation, Investigation, Methodology, Writing - original draft, Writing - review and editing; Alexander V Panfilov, Conceptualization, Supervision, Funding acquisition, Methodology, Writing - original draft, Writing - review and editing; Daniël A Pijnappels, Conceptualization, Resources, Supervision, Funding acquisition, Methodology, Writing - original draft, Project administration, Writing - review and editing

## Author ORCIDs
Rupamanjari Majumder (ID) https://orcid.org/0000-0002-3851-9225
Tim De Coster (ID) https://orcid.org/0000-0002-4942-9866
Arie O Verkerk (ID) http://orcid.org/0000-0003-2140-834X
Ronald Wilders (ID) http://orcid.org/0000-0002-1340-0869
Daniël A Pijnappels (ID) https://orcid.org/0000-0001-6731-4125

## Ethics
Human subjects: Conditional immortalization of human atrial myocytes was done with cells isolated from elective abortion material. Human tissue was obtained after individual permission using standard informed consent procedures. Experiments with these cells were performed in accordance with the national guidelines, approved by the Medical Ethical Committee of the Leiden University Medical Center (protocol P08.087), and conformed to the Declaration of Helsinki.

## Decision letter and Author response
Decision letter https://doi.org/10.7554/eLife.55921.sa1
Author response https://doi.org/10.7554/eLife.55921.sa2

# Additional files
## Supplementary files
• Source code 1. Key component of the graphics processing unit-usable code.

• Transparent reporting form

## Data availability
All data generated or analysed during this study are included in the manuscript and supporting files. Source data files have been provided for Figure 5.

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

## Appendix 1

We have designed two alternative schemes of an anti-arrhythmic gating mechanism that can also be used for a BioICD channel. Each scheme, shown as a Markov diagram, has its own strengths and weaknesses, as will be discussed below. We will denote them as Model II and Model III, and will refer to the model from the main text as Model I. BioICD Model I seems to be the easiest to engineer out of the three virtual BioICD channels, and therefore was selected for a more detailed study in 2D and 3D (see the Results section).

## Model II

In this model, we use a similar concept as in Model I, but with a different inactivation mechanism. The activation process is similar to Model I, with a closed state $C$ and an open state $O$. Inactivation occurs through a dedicated gate that has two states $I_O$ (open) and $I_C$ (closed). This gate is responsible for recovery of the channel after defibrillation and inactivates the current after $\approx$ 500 ms. Here too, the BioICD channel makes use of the 'deactivating agent' $a$, which in contrast to model I accumulates in the state $a_1$ during the diastolic phase (i.e. phase 4) of the AP. Thus, when the transmembrane potential of the cardiomyocyte is restored to its resting value, $a_1$ increases. This triggers a transition from the open state $O$ to the closed state $C$ (deactivation) with a decreased time constant, thereby leading to restoration of sinus rhythm. A schematic diagram of this model and its behaviour is presented in *Appendix 1—figure 1*. A detailed mathematical description of the state variables is provided in in *Equations 14, 15, 16, 17, 18, 19, 20, 21, 22, 23, 24, 25, 26*.

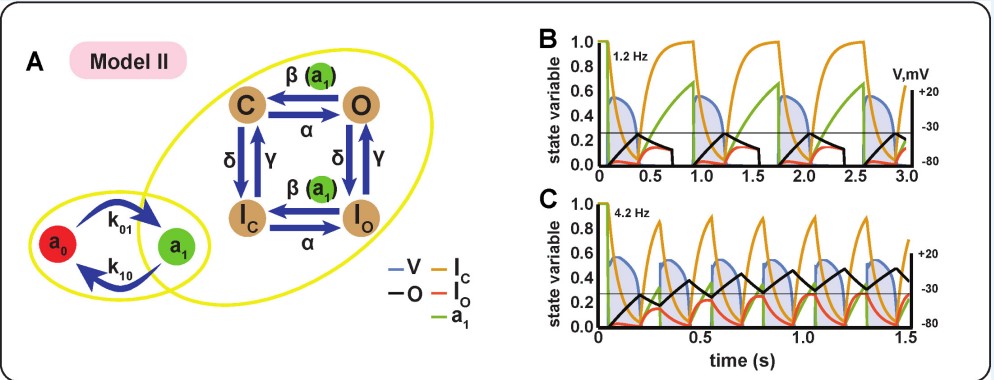

**Appendix 1—figure 1.** Schematic diagram and behaviour of BioICD channel Model II. (**A**) Markov diagram for BioICD Model II. (**B**) and (**C**) Traces of voltage ($V$) and state variables at 1.2 Hz and 4.2 Hz, respectively.

Total BioICD current is expressed as:

$$I_{BioICD} = G_{BioICD}\, I_O\, H(O - O_{thr})(V - E_{BioICD}). \tag{14}$$

Here, $G_{BioICD} = 10\,\text{nS/pF}$ is the maximal conductance of the whole collection of channels, $O_{thr} = 0.25$, $H$ is a Heaviside function, and $E_{BioICD} = 0\,\text{mV}$ is the reversal potential. Rate coefficients have explicitly been provided a dimensionality, while in Model I and III the kinetic coefficients are divided by a characteristic time.

$$\frac{\partial O}{\partial t} = \alpha C + \gamma I_O - (\beta + \delta)O, \tag{15}$$

$$\frac{\partial I_C}{\partial t} = \delta C + \beta I_O - (\alpha + \gamma)I_C, \tag{16}$$

$$\frac{\partial I_O}{\partial t} = \delta O + \alpha I_C - (\beta + \gamma)I_O, \tag{17}$$

where

$$\alpha = \begin{cases} 0\,\text{s}^{-1} & \text{if } V < -55.0\,\text{mV}, \\ 1\,\text{s}^{-1} & \text{otherwise}, \end{cases} \tag{18}$$

$$\beta = \begin{cases} 2\,\text{s}^{-1} & \text{if } V < -60.0\,\text{mV}, a_1 < a_{threshold}, \\ 1\,\text{ms}^{-1} & \text{if } V < -60.0\,\text{mV}, a_1 \geq a_{threshold}, \\ 1\,\text{s}^{-1} & \text{if } -60.0\,\text{mV} \leq V < -55.0\text{mV}, \\ 0^{-1} & \text{otherwise}. \end{cases} \tag{19}$$

$$O + C + I_O + I_C = 1, \tag{20}$$
$$a_0 + a_1 = 1. \tag{21}$$

In our 2D model, we used $a_{threshold} = 0.5$

$$\gamma = \begin{cases} 0\,\text{s}^{-1} & \text{if } V < -60.0\,\text{mV}, \\ 10^{-1} & \text{otherwise}, \end{cases} \tag{22}$$

$$\delta = \begin{cases} 10\,\text{s}^{-1} & \text{if } V < -60.0\,\text{mV}, \\ 0\,\text{s}^{-1} & \text{otherwise}. \end{cases} \tag{23}$$

Subunit $a$ has active ($a_1$) and inactive ($a_0$) states.

$$\frac{\partial a_1}{\partial t} = k_{01}a_0 - k_{10}a_1. \tag{24}$$

Transition rates between them are the following:

$$k_{01} = \frac{1.0}{(1.0 + e^{(V+60.0)/0.1})} \cdot \begin{cases} 2\,\text{s}^{-1}, & V < -60.0\,\text{mV} \\ 1\,\text{ms}^{-1}, & \text{otherwise}, \end{cases} \tag{25}$$

$$k_{10} = \frac{1.0}{(1.0 + e^{-(V+60.0)/0.1})} \cdot \begin{cases} 2\,\text{s}^{-1}, & V < -60.0\,\text{mV} \\ 1\,\text{ms}^{-1}, & \text{otherwise}. \end{cases} \tag{26}$$

## Model III

This model comprises six identical subunits with four states each (see **Appendix 1—figure 2A**): $I$ (inactive state), $C_1$ (1st closed state), $C_2$ (2nd closed state) and $O$ (open state). The mechanism is less sensitive to the changes in AP shape or duration by design, but requires more states in a Markov chain model.

Initially, all subunits are in the first closed state $C_1$. When the cell depolarizes ($V > -25\ mV$), most of the subunits of the channel transit to the inactive state $I$, whereas the remaining fraction of subunits change conformation to the other closed state $C_2$. The number of subunits in the second closed state ($C_2$) always increases approximately to the same value after each upstroke (see green spikes of $C_2$ in **Appendix 1—figure 2B and C**). The channel can then open from the $C_2$ state, but not from $C_1$ or $I$. The opening and inactivation from the $C_2$ state occurs slowly and independent of the voltage ($k_4, k_5$), which slows down the overall kinetics of the channel. Therefore, the number of open channels ($O$) reflects the average/integrated number of upstrokes in a unit of time (see **Appendix 1—figure 2B and C**, black lines). The

channels recover from inactivation mainly during the late phase of the AP, when the transmembrane voltage ranges from −60 mV to −25 mV (as can be seen in **Appendix 1—figure 2B and C** by an increase of $C_1$ during the final phase of the AP). At high frequencies, the number of channels in the second closed state $C_2$ and in the open state $O$ increases. The channels with all subunits in the open state contribute to the defibrillation current. After arrhythmia termination, in the absence of new upstrokes, the subunits of the channels recover fast to the first closed state $C_1$ (see **Appendix 1—figure 2B and C**, orange line). The Markov diagram for this model is shown in **Appendix 1—figure 2A**. A detailed mathematical description of the state variables is provided in **Equations 27, 28, 29, 30, 31**).

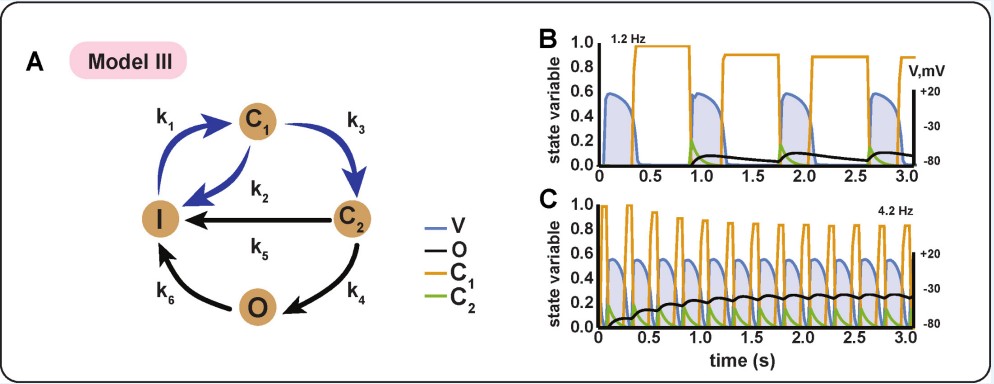

**Appendix 1—figure 2.** Schematic diagram and behaviour of BioICD channel Model III. (**A**) Markov diagram for BioICD Model III. (**B**) and (**C**) Traces of voltage ($V$) and state variables at 1.2 Hz and 4.2 Hz, respectively.

Total BioICD current is expressed as:

$$I_{BioICD} = G_{BioICD}O^6(V - E_{BioICD}), \tag{27}$$

where $G_{BioICD} = 1000\,\text{nS/pF}$ and $E_{BioICD} = 0\,\text{mV}$.

$$\frac{\partial C_1}{\partial t} = \frac{k_1 I - k_2 C_1 - k_3 C_1}{\tau}, \tag{28}$$

$$\frac{\partial C_2}{\partial t} = \frac{k_3 C_1 - k_4 C_2 - k_5 C_2}{\tau}, \tag{29}$$

$$\frac{\partial O}{\partial t} = \frac{k_4 C_2 - k_6 O}{\tau}, \tag{30}$$

$$I + C_1 + C_2 + O = 1, \tag{31}$$

where $\tau = 7\,\text{ms}$ and $k_1$, $k_2$ and $k_3$ are functions of voltage:

$$k_1 = \begin{cases} 1.0 & \text{if } \textit{-60}\text{ mV} < V < \text{-25 mV}, \\ 0.0 & \text{otherwise}, \end{cases} \tag{32}$$

$$k_2 = \begin{cases} 30.0 & \text{if } V > \text{-25 mV}, \\ 0.0 & \text{otherwise}, \end{cases} \quad k_3 = \begin{cases} 10.0 & \text{if } V > \text{-25 mV}, \\ 0.0 & \text{otherwise}. \end{cases} \tag{33}$$

Other kinetic coefficients are constant: $k_4 = 0.05$, $k_5 = 0.01$, and $k_6 = 0.05$.

Finally, in **Appendix 1—figure 3** we demonstrate the operation of our BioICD Model II and Model III in simulated homogeneous human ventricular monolayers. In each case, the channel remains closed during sinus rhythm. However, upon arrhythmia initiation, such abnormal electrical activity is terminated within 1.2 s by transient activation of the BioICD channel resulting in the restoration of sinus rhythm.

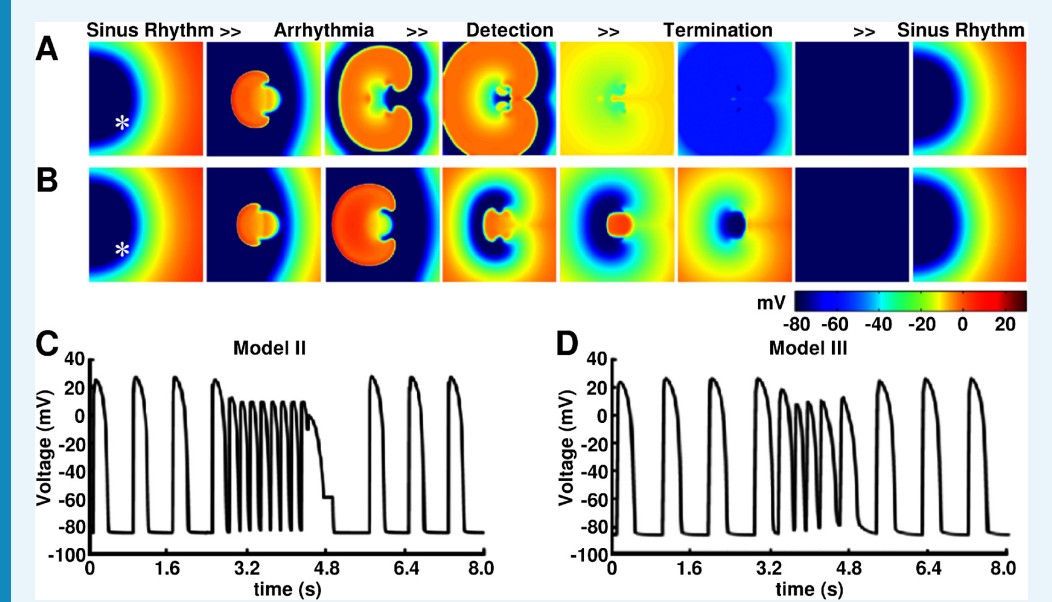

**Appendix 1—figure 3.** Anti-arrhythmic action of BioICD Model II and III in monolayers of human ventricular tissue (TNNP model). (**A** and **B**) Pseudocolor plots of subsequent time frames of the process of biological auto-detection and termination of reentrant tachyarrhythmias, followed by restoration of sinus rhythm, as demonstrated by using BioICD Model II and Model III, respectively. (**C** and **D**) Voltage traces from representative cardiomyocytes (white asterisks) in the simulation domains for Model II and Model III, respectively.

