## [Decision Letter]

**Acceptance summary:**

This manuscript suggests that an engineered ion channel can act as a novel feedback mechanism to protect against arrhythmias. Evidence is provided that a channel with appropriate kinetics can do this in computer models, and using a dynamic clamp, in atrial myocytes.

**Decision letter after peer review:**

Thank you for submitting your article "Self-restoration of cardiac excitation rhythm by anti-arrhythmic ion channel gating" for consideration by *eLife*. Your article has been reviewed by Kenton Swartz as the Senior Editor, a Reviewing Editor, and two reviewers. The following individual involved in review of your submission have agreed to reveal their identity: Gil Bub (Reviewer #2).

The reviewers have discussed the reviews with one another and the Reviewing Editor has drafted this decision to help you prepare a revised submission. In recognition of the fact that revisions may take longer we typically allow, until the research enterprise restarts in full, we will give authors as much time as they need to submit revised manuscripts.

Summary:

This manuscript suggests that an engineered channel can act as a novel feedback system to protect against arrhythmias. It is shown that a channel with appropriate kinetics can do this in computer models and, using the dynamic clamp, in atrial myocytes. The idea and modeling are fascinating. There are, of course, two problems in potential implementation. (1) How will an appropriate channel be designed and synthesized? (2) The channel would need to be expressed long term throughout the ventricle in order to deal with ventricular arrhythmias – no easy feat.

Essential revisions:

1) The experiments in the new Figure 5, while compelling, potentially can benefit from some clarification. As I understand it, the clamp protocol injects current based on a measurement of the current voltage. In a real channel, the current may be limited and offset by compensating currents from other channels, while here this may be overpowered the clamp. If the latter is true, then the experiment doesn't show that a BioICD channel would work in real life (the heart cell being in effect forced to show the right result by the clamp). I concede that this point may turn out to be a misunderstanding on my part as I have not worked with voltage or current clamp protocols (in which case the authors can answer this in rebuttal without changes to the manuscript).

2) It is unclear how the BioICD channel would operate under conditions of sustained exercise. It would be very helpful if the authors would provide both dynamic and standard restitution plots (up to 3.3 Hz, a good upper limit for heart rate in a healthy adult) to show that the channel wouldn't interfere with cardiac function in normal circumstances.

3) Heterogeneous expression of a BioICD channel may result in increased APD dispersion during exercise (e.g. the patient goes jogging). It would be good to repeat the simulations suggested in point 2 above for cells with a 10% difference in BioICD channel concentration to make sure that this overall approach is not potentially pro-arrhythmic. Of course, if the restitution curves in point 2 are identical to those of a cell without a BioICD channel then this simulation is not necessary.

---

## [Author Response]

Summary:This manuscript suggests that an engineered channel can act as a novel feedback system to protect against arrhythmias. It is shown that a channel with appropriate kinetics can do this in computer models and, using the dynamic clamp, in atrial myocytes. The idea and modeling are fascinating. There are, of course, two problems in potential implementation. (1) How will an appropriate channel be designed and synthesized? (2) The channel would need to be expressed long term throughout the ventricle in order to deal with ventricular arrhythmias – no easy feat.

We agree that the engineering and expression of an ion channel enabling self-restoration of cardiac excitation rhythm in sustained arrhythmias (i.e. BioICD channel) is a challenging but fascinating road ahead of us. As rightfully remarked by the editors and reviewers, there are indeed several hurdles to be taken regarding the implementation of a BioICD ion channel. However, in the Discussion section of our paper, we explain that the essential techniques for designing, synthesizing and incorporating such a channel are already present to further explore the realization of BioICD channels for the heart. Regarding the long-term expression of these channels throughout the heart, more research would certainly be needed. Experimentally, we will address these important aspects and hope that this proof-of-concept paper will act as a source of inspiration and as incentive for other groups to join the further exploration of the biological defibrillation concept. We know already from pre-clinical studies that with current viral gene transfer technology, cardiomyocyte transduction rates of close to 100% can be achieved in adult rat atria and ventricles (Nyns et al., 2016, 2019) and in adult pig atria (Kikuchi et al., 2005). It remains to be studied whether safe and effective self-restoration of cardiac excitation rhythm by BioICD channels requires such high transduction rates throughout cardiac tissue. Regarding the longevity of transgene expression, it is known that current viral vector technology potentially allows for many years (i.e. chronic) of transgene expression in the heart (Rincon et al., 2015). Again, more dedicated research is needed, but the tools and methods are present to further optimize long-term expression of transgenes in the heart. These facts make us confident that the implementation of BioICD is within the realms of possibilities, although it may indeed not be an easy feat to accomplish, but worth the effort given the unique and much desired advantages over conventional non-biological anti-arrhythmic strategies.

Essential revisions:1) The experiments in the new Figure 5, while compelling, potentially can benefit from some clarification. As I understand it, the clamp protocol injects current based on a measurement of the current voltage. In a real channel, the current may be limited and offset by compensating currents from other channels, while here this may be overpowered the clamp. If the latter is true, then the experiment doesn't show that a BioICD channel would work in real life (the heart cell being in effect forced to show the right result by the clamp). I concede that this point may turn out to be a misunderstanding on my part as I have not worked with voltage or current clamp protocols (in which case the authors can answer this in rebuttal without changes to the manuscript).

We may have been unclear in explaining the workings of dynamic patch clamp. Please let us clarify. As mentioned by the reviewer, endogenous ionic currents do add to each other and to the BioICD current. Ionic currents interact with each other primarily via the membrane potential because the current amplitude is a function of the driving force (*V* − *E_rev_*) at all times. So, while each current has an effect on the membrane potential, in turn the membrane potential has an effect on the current. This behaviour was incorporated in the BioICD current definition (Materials and methods section Equation 1) and actually installed a BioICD conductance as an extra parallel conductance in the whole cell. Thus, the dynamic clamp experiment mimicked the voltage-current interaction, since the membrane potential was only recorded, and was allowed to move freely as a consequence of the net activity of all endogenous ion channels and the BioICD current during the entire experiment. The BioICD current amplitude was dependent on the actual membrane potential at all times, according to the model definitions, therefore interacting with the endogenous currents in the same way the endogenous currents interact with each other. We have now rewritten the text piece about the dynamic clamp experiment for clarification, please see subsection “Self-restoration of excitation rhythm in human atrial myocytes”:

“In order to assess, in actual cardiomyocytes, the anti-arrhythmic effect of the BioICD channel, the BioICD current was mimicked in vitro using dynamic patch-clamping (Wilders, 2006). […] A real-time interface between the patch-clamp amplifier and a computer constituted the feedback loop (Figure 5B) that allowed us to mimic the presence of the BioICD channel, and to observe the effects of BioICD channel activity in cardiomyocytes (Figure 5C-D).”

2) It is unclear how the BioICD channel would operate under conditions of sustained exercise. It would be very helpful if the authors would provide both dynamic and standard restitution plots (up to 3.3 Hz, a good upper limit for heart rate in a healthy adult) to show that the channel wouldn't interfere with cardiac function in normal circumstances.

We thank the reviewer for his insightful comment. First, we would like to make a case for not including the standard restitution plot. Since BioICD channel activity has to develop over time, it will not activate after a sudden/short-lasting change in heart rate, but rather with sustained high rhythm (see Figure 3E-H). Therefore, the standard restitution plots of the model with or without BioICD will lie on top of each other. Differences can be observed when changes of longer duration are allowed and we let the action potentials settle, as is the case with a dynamic protocol. We therefore produced the relevant dynamic restitution plots. With the obtained data we updated Figure 3 from the main manuscript.

Figure 3J shows the dynamic restitution curve (APD_90_) for a human ventricular myocyte. The restitution curve for the model without a BioICD channel (orange curve) corresponds to the one that can be found in the original paper by tenTusscher and Panfilov (2006) with the parameter set corresponding to slope 1.1 (Figure 5F in tenTusscher and Panfilov, 2006). In black, the restitution curve is shown when the BioICD channel is present. This curve runs nicely on top of the orange one, except for small diastolic intervals. At these small diastolic intervals, the BioICD current activates. The points where activation was present are kept out of the restitution curve. In the short run-up to activation, a small deviation can be seen with regards to the orange curve (max. 10 ms).

For two chosen frequencies (1.2 Hz and 2.8 Hz), action potential traces are shown in Figure 3I. The corresponding points in the restitution curve are indicated as well in Figure 3J. It can be seen that the small effect of the BioICD is mainly visible in the tail of the action potential.

Currently, the BioICD current activates at a frequency of 2.9 Hz (174 bpm). This is lower than the 3.3 Hz (198 bpm) that was asked for by the reviewers. However, the gating of the BioICD channel was designed to target persisting hazardous arrhythmias. Some of these arrhythmias already occur at frequencies lower than 3.3 Hz (Roy et al., 1982; Naccarelli et al., 1983). This creates a rationale for designing a BioICD channel that should activate at lower frequencies than the upper limit for heart rate in a healthy adult. This may not necessarily create a problem because the average patient suffering from sustained arrhythmias tends not to physically challenge him or herself to the extent that the heart rate approaches its upper limit. However, as mentioned in the Discussion section, in case one would wish to change the frequency at which the ion channel activates (limiting or increasing maximal heart rate, but thereby increasing or limiting the restoring function of the BioICD channel), the opening and closing rates of the ion channel can be changed.

Besides the new Figures, the following text has been included in the manuscript (subsection “Self-restoration of excitation rhythm in human atrial myocytes”):

“As shown in Figure 3I, the presence of the BioICD channel exerts no significant influence on the AP at sinus rhythm (1.2 Hz) (the black and orange traces coincide), but slightly increases the APD at close-to-arrhythmic frequencies. This is also accounted for in the APD restitution curve of Figure 3J, which shows that the BioICD channel effectively increases the minimal APD of the cardiomyocytes. As a consequence of their sensitivity to frequency, the BioICD channels are unable to detect and eliminate reentrant activity anchored to scars if the activation frequency is too low, for example in substrates with large scars (see Video 5 in the supporting information).”

3) Heterogeneous expression of a BioICD channel may result in increased APD dispersion during exercise (e.g. the patient goes jogging). It would be good to repeat the simulations suggested in point 2 above for cells with a 10% difference in BioICD channel concentration to make sure that this overall approach is not potentially pro-arrhythmic. Of course, if the restitution curves in point 2 are identical to those of a cell without a BioICD channel then this simulation is not necessary.

Although the simulations that were carried out to obtain the restitution curve for answering point 2 showed that there is a negligible difference between both curves, we still carried out the suggested simulations with 90 and 110% BioICD channel concentration. From Author response image 1 it can be seen that all the curves (90, 100 and 110%) lie on top of each other, except at the very end where we are dealing with high frequencies and the BioICD current almost activates (APD differences of 1 to 2 ms can be observed between the 3 BioICD curves, and about 10 ms with respect to the APD of a regular cell). As an extra check, a CV restitution curve was generated (Author response image 1), where it can be seen that also these curves lie on top of each other. This indicates that the anti-arrhythmic effect of BioICD does not necessarily require the same concentration of ion channel in each and every cell. Given the outcome of the dynamic restitution analyses and the comment of the reviewers regarding the necessity of this particular simulation, we decided to not include these data in the manuscript.

**Author response image 1. respfig1:** 1D cable action potential duration (APD) restitution. (**A**) Dynamic APD restitution curve for the original parameters of the human ventricular cardiomyocyte model, with and without BioICD channel. (**B**) Conduction velocity restitution curve for the TNNP model (tenTusscher and Panfilov, 2006) with and without BioICD channel for different channel densities. For all three channel densities (90, 100 and 110% of the model described in the main manuscript), the curves lie on top of each other.

References

Kikuchi, K., McDonald, A. D., Sasano, T., and Donahue, J. K. (2005). Targeted modifcation of atrial electrophysiology by homogeneous transmural atrial gene transfer. Circulation, 111(3):264-270.

Naccarelli, G. V., Zipes, D. P., Rahilly, G. T., Heger, J. J., and Prystowsky, E. N. (1983). Influence of tachycardia cycle length and antiarrhythmic drugs on pacing termination and acceleration of ventricular tachycardia. American heart journal, 105(1):1-5.

Rincon, M. Y., VandenDriessche, T., and Chuah, M. K. (2015). Gene therapy for cardiovascular disease: advances in vector development, targeting, and delivery for clinical translation. Cardiovascular research, 108(1):4-20.

Roy, D., Waxman, H. L., Buxton, A. E., Marchlinski, F. E., Cain, M. E., Gardner, M. J., and Josephson, M. E. (1982). Termination of ventricular tachycardia: role of tachycardia cycle length. The American journal of cardiology, 50(6):1346-1350.